# Dissemination of trial results to participants in phase III pragmatic clinical trials: an audit of trial investigators intentions

M Zulfiqar Raza, Hanne Bruhn, Katie Gillies 

Health Services Research Unit, University of Aberdeen, Aberdeen, UK

**Correspondence to**
Dr Katie Gillies;
k.gillies@abdn.ac.uk

## ABSTRACT

**Objective** To determine the proportion of Phase III clinical trials given a favourable opinion by a research ethics committee in the UK that provided trial results to those who participated.

**Design** Audit of records.

**Setting** Phase III clinical trials registered on the UK's research permissions system (Integrated Research Application System) between the 1 January 2012 to 31 December 2017.

**Main outcome measures** Proportion of trial investigators that intended to provide results to trial participants compared against what trials reported to ethics committees at the end of study.

**Results** Out of 1404 Phase III trials, 87.7% (n=1231) trials stated they intended to disseminate results to participants while 12.3% (n=173) trials stated they would not. Out of these 1231 trials, 18.8% (n=231) trials intended to actively communicate trial results or a means of accessing results to their participants, a further 80.5% (n=991) reported passive intention to disseminate and for the remainder (n=9) the process was unclear. Of the 370 End of Study reports (30% of all included studies) that could be accessed 10 (2.7%) explicitly mentioned activities related to dissemination of findings to participants with the majority (74.9%) having no mention and a further 22.4% of reports not being accessible. Of the 10 which did report dissemination of results to participants the majority (n=6) were through a lay summary or letter.

**Conclusions** Reported intention to disseminate results to trial participants among trial investigators is high, however, reporting of feedback methods is lacking. In addition, mechanisms to ensure intentions to disseminate trial results are translated into actual behaviour need to be put in place to ensure those who participate in trials have the opportunity to find out about the results.

## INTRODUCTION

Clinical trials and research are increasing in the UK. In 2018, a total of 870 250 participants took part in National Institute for Health Research Clinical Research Network supported clinical research studies in England alone — marking an increase of over 140 000 over the previous year.[1] The cumulative cost of these studies was around

### Strengths and limitations of this study

► First audit of the Integrated Research Application System (IRAS) to investigate trial investigators reported intention to disseminate trial results to participants.

► Describes frequency of intention to disseminate and reported plans for dissemination.

► Links End of Study reports to original IRAS applications and provides a summary of overall behaviours about reporting of dissemination of results in said reports.

► Linkage with End of Study reports to report actual behaviour regarding dissemination is limited due to no explicit requirement from Health Research Authority to report this activity in final report.

£6 billion and is likely to increase as the National Health Service long-term plan targets to include one million people taking part in research by 2023/2024.[1] This increase in participants numbers has the potential to translate into significant improvement in delivery of healthcare as long as findings are disseminated to those with responsibility to make a change (policymakers) and the end users of the services (patients and healthcare professionals).

In 2008, a key review based on 28 empirical studies demonstrated that 90% of participants would want to be informed of the results of the research that they were involved in.[2] Despite this interest shown by participants, little is being done to provide them with results.[3] A survey on research participant experience showed that 90% of respondents were happy with the information that they received before or during the research. However, there was little indication that they were provided with or made aware of the opportunity to access results after completion.[4] This lack of attention to meeting expectations of research participants is not acceptable.

When aligned with recent initiatives to improve research integrity through ensuring trials are registered and that their results are published, it seems an obvious next step to make sure those who participated in them (and who which without they would not be possible) are informed of the results.

In order to encourage the dissemination of results, the Health Research Authority (HRA, whose core purpose is 'to protect and promote the interests of patients and public in health and social care research' in the UK) published guidelines, recommending that all researchers communicate results to their study participants and at the very least offer the results.[5] The guidelines also recommend patient and public involvement (PPI) in all aspects of the research process.[5] This refers to the involvement of patients and/or members of the public in the design or undertaking of the research process.[6] An example of this would be patient input regarding the mode of dissemination of results in order to improve the feedback process. These contributions can be very valuable as they can provide an alternative perspective that the researchers may not have considered and can ensure the materials are accessible to non-experts. Unlike the dissemination of results, which is not mandatory for Phase III trials, the inclusion of PPI in the research process is mandated by funding bodies as a prerequisite to obtaining funding.

In the UK, applications for ethical approval are made through the Integrated Research Application System (IRAS). The IRAS form includes questions regarding the researchers' intention to disseminate results to participants as well as any intended PPI. On completion, the research team must then submit a declaration of end of study to the research ethics committee (REC) followed by a final ethics report within 12 months of the completion of the study (the End of Study report). The final ethics report should confirm any steps taken to disseminate results to participants.[7] The guidelines also instruct researchers about the information to be included in the patient's end of study information sheet, which should as a minimum offer the results and specify when and how participants should expect to receive results.

In addition, this national level guidance, there is international recognition of the ethical imperative (specified within the Declaration of Helsinki) to offer results which is represented by the statement 'all medical research subjects should be given the option of being informed about the general outcome and results of the study'.[8]

While there is a need to provide results of any research study which a participant has contributed to, providing results from clinical trials has salience in the current research transparency landscape.[9] Phase III clinical trials also hold a position of particular importance given all participants will have had no choice in the treatment they received, many may not know what intervention they received, several will have provided data through patient reported outcomes and many are publicly funded. At the very least, trial teams should be making the results of the

studies to which these individuals contribute to available and accessible to them in appropriate ways.

This study aims to assess whether researchers in the UK intend to inform participants of the trial results, plans for how results are provided, how patients are involved in this process and finally, whether those trial teams that intended to provide results report this activity in their End of Study reports.

## METHODS
### Inclusion criteria
This study included all applications on the IRAS during the period 1 January 2012 to 31 December 2017 where the research team had selected filter question (defining the work as a clinical trial) and that had received a favourable REC opinion and carried out in the UK. IRAS is the UK's online system for the permissions and approvals for health, social and community care research.

### Data request and extraction
Information regarding the IRAS form submitted by researchers was requested from the HRA. Specific data on study descriptors such as: IRAS Project ID; REC name; REC reference; Study title; Protocol version and date; etc, was requested. In addition, data from project relevant questions, relating to patient and public involvement, plans for dissemination and whether participants would receive results, from within the IRAS form were requested, (see Box 1 for the specific questions and the data types contained within them). On receipt of the data from IRAS, additional criteria were applied to select Phase III clinical trials for inclusion (filtering to select only those studies that reported 'Yes' to the filter 'Therapeutic confirmatory trial (Phase III)'. The rationale for only including Phase III randomised controlled trials in this audit was due to Phase III trials largely collecting patient-reported outcomes, for which there may be more potential for demonstrable change in practice and as such greater buy in from participants to receive the overall results from the data collected.

The data items requested from IRAS were specified in the 'HARP Software Change/Management Information Request Form'. In addition to the data contained within

---

**Box 1 Requested filter questions from IRAS form**

A14-1. In which aspects of the research process have you actively involved, or will involve, patients, service users, and/or their carers, or members of the public? Give details of involvement, or if none please justify the absence of involvement.
► Data provided: nominal data and open-ended text
A51. How do you intend to report and disseminate the results of the study?
► Data provided: nominal data and open-ended text
A53. Will you inform participants of the results? Please give details of how you will inform participants or justify if not doing so.
► Data provided: dichotomous data (yes/no) and open-ended text

---

IRAS we also requested access to final ethics reports (ie, the End of Study reports) through the HRA Assessment Review Portal (HARP) in order to confirm a match between information provided in the IRAS form about what was planned for feeding back results to participants with what actually happened, as reported in the final report. Final reports were identified by searching for specific REC reference identification numbers within HARP.

## Data analysis

IRAS responses which provided nominal data were summarised using descriptive statistics such as frequencies and percentages for example, 'Will you inform participants of the results? – Yes/No?'. Free text responses were categorised using content analysis. Single level coding was applied, codes were developed iteratively in discussion between team members and coding was performed by one team member. In order to assess the involvement of trial teams in the act of dissemination, our team categorised the intended means of dissemination of results as being either active or passive:

► Active: The trial team directly informed participants of the means by which the results could be accessed. For example, by a letter or by including a web link to the results.
► Passive: Trial team did not directly inform participants of a means to access trial results. For example, where the responsibility to forward the results was placed on the site team.

## Patient and public involvement

This audit forms part of a larger project that aims to develop recommendations for researchers on how to report clinical trial results appropriately to participants (RECAP: researchregistry4085). There are two patient partners on the Advisory group for the RECAP project who have contributed to this substudy through discussion of results at team meetings. In addition, the HRA patient and public involvement lead has also been involved in conversations about this audit and had opportunity to comment and guide interpretation of the results in advance of final analysis.

## RESULTS
### Data mining

Data on a total of 6826 trials (which had received a favourable opinion from a REC) was received in the initial data set collated by the HRA based on the requested filter questions. One thousand four hundred and four of these were identified as Phase III trials as prespecified by the trial team on the IRAS system (studies that reported 'Yes' to the filter 'Therapeutic confirmatory trial (Phase III) – figure 1).

### Intention to disseminate

A total of 1231 (87.7%) trial teams stated they intended to disseminate results to participants while 173 (12.3%)

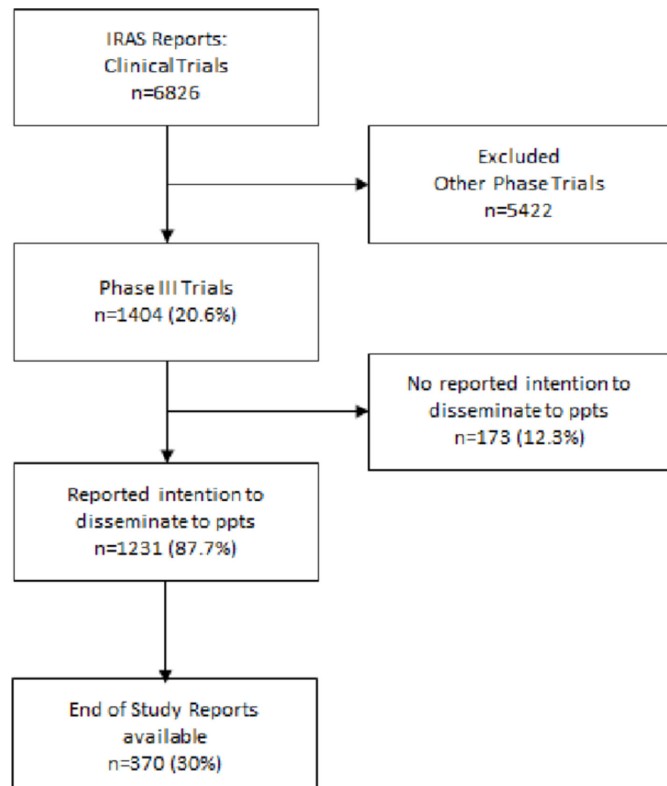

**Figure 1** Summary of search results.

trials stated they would not. Researchers were then asked to provide details on how they intended to do so. Of those that said yes, we identified 231 (18.8%) as reporting an active effort to disseminate results, that is, the trial team/sponsor actively made arrangements to provide participants with results. The most commonly reported mode of active dissemination was directing participants to a website with results (see table 1). This was reported in 74 (32%) of trials that planned to actively disseminate to participants. Some trials included this at the beginning of the trial in the Participant Information Leaflet or at the end of participants involvement in the End of Study information sheet. Fifty-three (22.9%) trial teams stated that they intended to provide either lay summaries or information sheets but did not specify any other information. Forty-four (19%) intended to send the results by mail directly to the participants. The 'other' category involves 4 (1.7%) trials that included reasons such as organising 'dissemination events' and holding meetings with the participants.

Within the trials whose teams reported an intention to disseminate results to participants, we coded 991 as reporting a passive method to disseminate results that is, there were no formal arrangements made to provide patients with access to the results. The most common method of passive dissemination stated the local site team, such as the study doctors or trial investigators, would provide results at their discretion. This accounted for 549 (55.4%) trials. Another 339 (34.2%) trial teams stated that results would be provided on request but did

**Table 1** Summary of trial team responses regarding intention to disseminate

| Means of dissemination | Intention to disseminate results to participants | |
| --- | --- | --- |
| | Yes | No |
| **Active** | | |
| Provision of web link to results | 74 (32%) | – |
| Postal letter | 44 (19%) | – |
| Patient information sheet | 37 (16%) | – |
| Clinic appointment | 23 (10%) | – |
| Lay patient summary | 16 (6.9%) | – |
| Patient choice of mode of delivery | 16 (6.9%) | – |
| PPI group | 10 (4.3%) | – |
| Newsletter | 5 (2.2%) | – |
| Email | 2 (0.9%) | – |
| Other (eg, face-to-face meetings) | 4 (1.7%) | – |
| *Total* | *231 (18.8%)* | - |
| **Passive** | | |
| Trial linked staff (eg, discretion of study doctor) | 549 (55.4%) | 72 (41.6%) |
| Participant initiated request | 339 (34.2%) | 49 (28.3%) |
| Public domain (trial website) | 84 (8.5%) | 24 (13.9%) |
| Conference/scientific publication | 17 (1.7%) | – |
| Media | 1 (0.1%) | – |
| Public representative meeting | 1 (0.1%) | – |
| No reason stated | – | 24 (13.9%) |
| Other | – | 4 (2.3%) |
| *Total* | *991 (80.5%)* | *173 (100%)* |
| **Unclear** | 9 (0.7%) | - |
| *TOTAL* | *1231* | *173* |

PPI, patient and public involvement.

not specify how the participants would be given the opportunity to request results. Finally, 84 (8.5%) intended to make the results available in the public domain but did not specify how the participants would be informed of or directed to these results.

Responses coded as 'Unclear' (of which there were 9, 0.7%) either left the question unanswered or provided a vague statement. For example, 'Participants will be informed of the results post-study.'

A total of 173 (12.3%) trial teams reported that they did not intend to provide participants with the results. Of these, 72 (41.6%) stated that there were no plans to disseminate results, but that study investigators or study doctors may pass on the results. Forty-nine (28.3%) stated that results would be provided if the participants expressed an interest or requested them. Twenty-four (13.9%) provided no reason. Another 24 (13.9%) stated that results would be made available in the public domain but not sent to participants directly. 'Other' (n=4, 2.3%) includes trials that mentioned the use of patient groups to disseminate results or provided non-specific statements

such as 'Patients will be informed about the results of their study in an individual manner'.

### Patient and public involvement
We also wanted to determine whether those trials that planned to disseminate results to trial participants were better overall at including patients in the design and conduct of the trial. Therefore, we analysed whether and how the trial teams that intended to disseminate results included patients as partners in their studies.

Within the sample of 1231 who planned to disseminate results to participants, 381 (31%) trial teams also reported they intended to involve patients or the public in the design or conduct of their trial. The largest proportion of PPI was observed in the dissemination phase with 227 trials accounting for 59.6% while only 4.5% (n=17) of trial teams proposed input during the analysis phase. Elsewhere, 180 (47.2%) reported they would incorporate PPI in the design phase of the research; 123 (32.3%) would seek input while the undertaking of the trial and 121 (31.7%) proposed to involve patients or public in the

**Table 2** Researcher-reported patient and public involvement in trial design and/or conduct

| | IRAS-reported PPI in design and/or conduct of trial | | | |
| | Intention to disseminate results to participants (n=381) | | No intention to disseminate results to participants (n=42) | |
| Aspect of trial | *Frequency* | % | *Frequency* | % |
|---|---|---|---|---|
| Design | 180 | 47.2 | 16 | 38.1 |
| Management | 121 | 31.7 | 6 | 14.3 |
| Undertaking | 123 | 32.3 | 28 | 66.7 |
| Analysis | 17 | 4.5 | 2 | 4.8 |
| Dissemination | 227 | 59.6 | 11 | 26.2 |

*Totals for % are greater than 100 as categories are not mutually exclusive and research teams could report PPI across several aspects of the research.
IRAS, Integrated Research Application System; PPI, patient and public involvement.

management phase (see table 2). It is important to note that involvement was not mutually exclusive to one individual design or conduct category and researchers could select involvement across multiple categories.

Forty-two (24%) of the 173 trial teams that had no intention of disseminating results back to participants did report patient or public involvement in at least one aspect of the trial process. Within this sample of 42, undertaking the research was reported most frequently as involving PPI (n=28, 66.6%), followed by design (n=16, 38.1%), dissemination (n=11, 26.2%), management (n=6, 14.3%) and analysis (n=2, 4.8%). Again, it is important to note that involvement was not mutually exclusive to one category.

A total of 850 (69%), from the 1231 trial teams that intended to disseminate result to participants, stated they would not be involving PPI partners at any stage of the research process. Among these, 244 (28.7%) deemed PPI to be unnecessary due to the sufficient expertise present among the members of the research team or other sources for example, 'It was felt that sufficient input had been gained from other sources'. A further 213 (25.1%) trials responded that it was inappropriate to involve members of the public due to the complex or experimental nature of the trial or the use of an unlicensed drug. One hundred and forty-five (17.1%) trials stated that all aspects of the research process were sole responsibility of the trial sponsor. One hundred and five (12.4%) trials did not provide an explanation for not doing so. 'Other' involved 34 (3.9%) trials that do not give a specific reason for the lack of PPI or simply describe the details of the trial itself. For example, 'no patients, services and/or their carers or members of the public were involved with the design of the protocol'. Finally, 'Prescribed design' accounted for 12 (1.4%) of the responses, which refers to studies that are using previously implemented trial designs and who deemed PPI not necessary. Responses are summarised in table 3.

### End of study report
Data for the 1231 trial teams that intended to disseminate results was extracted from HARP to identify, first, whether these studies submitted an End of Study report. A large proportion of trials (n=517 (42%)) were still in progress when the data was requested while 90 trials (7.3%) had been terminated or abandoned and as such no End of Study reports were available for these trials. Of the 624 completed trials, 370 (59.3% of completed trials and 30.1% total sample) submitted a final ethics report, while 127 (20.4% of completed trials and 10.3% of total sample) failed to do so and 127 (20.4% of completed trials and 10.3% of total sample) had incomplete data registered within the HARP system making analysis difficult. (table 4).

Of the 370 studies that did submit an End of Study report, the majority of the trial teams (n=277, 74.9%) did not mention any arrangements made regarding the dissemination of trial results back to participants yet all expressed intention to do so on the original IRAS application. Six studies (1.6%) provided a copy of the lay summary or referred to it in the report or the cover letter.

**Table 3** Summary of responses to justification of no patient and public involvement

| Reason | Frequency (n) | % of total |
|---|---|---|
| Sufficient expertise | 244 | 28.7 |
| Sponsor responsibility | 145 | 17.1 |
| Inappropriate – experimental nature of trial | 110 | 12.9 |
| Unanswered | 105 | 12.4 |
| Inappropriate – complexity of trial | 83 | 9.8 |
| Commercial trial | 82 | 9.6 |
| Inappropriate – unprescribed drug | 20 | 2.4 |
| Confidentiality | 15 | 1.8 |
| Prescribed research design | 12 | 1.4 |
| Other | 34 | 3.9 |
| *TOTALS* | *850* | *100* |

**Table 4** End of study report status

| Report status | | Frequency (n) | % of total |
|---|---|---|---|
| Completed trials | Submitted | 370 | 30.0 |
| | Not submitted | 127 | 10.3 |
| | Incomplete HARP data* | 127 | 10.3 |
| Trial in progress† | | 517 | 42.0 |
| Trial terminated/abandoned | | 90 | 7.3 |
| *TOTAL* | | *1231* | *100* |

*Incomplete HARP data: HARP has trial registered but is incomplete. For example, does not clearly state that trial ever started/little or no documentation uploaded to HARP.
†Trial in progress: trial currently recruiting or in follow-up, or, not yet started, or, trial complete and not reported but has up to 12 months to report.
HARP, HRA Assessment Review Portal; HRA, Health Research Authority.

Evidence of other strategies used to inform participants of the trial results were also poorly represented with 2 (0.5%) studies providing the patient End of Study sheet, 1 (0.3%) offering a final follow-up visit, and another 1 (0.3%) mentioning presentation at a scientific conference. While indicating the End of Study reports had been uploaded, the reports of 83 (22.4%) trials were inaccessible due to some requiring passwords or email access or yet to be uploaded by the REC to the HARP system. Therefore, details from these reports could not be extracted or included in the analysis, see table 5.

## DISCUSSION
### Key findings
This study reports the first audit of researcher intentions and self-reported behaviours with regard to dissemination of clinical trial results to participants across the UK using reports within the HRA regulatory system. We have found that while the majority (n=1231, 87.7%) of trial teams stated in their applications that they intended to disseminate trial results to the participant, less than 20% (n=231, 18.8%) specified some form of direct 'active' communication with their participants. The majority of trial teams (80.5%) left the responsibility of participants accessing trial results with the clinical care team or on the participant themselves. The other key finding relates to the dissemination behaviour reported by trial teams in their End of Study report, which demonstrated that 59.3% of completed trials had submitted an End of Study report compared with 20.4% that had not. However, the majority (74.9%) of End of Study reports did not mention any arrangements for the provision of trial results to participants.

The findings from our study show the potential variability in reporting trial results back to participants with many trial teams not doing so, which is in line with findings from previous studies.[10] Also, the variability we identified with regard to how the results would be provided (ie, paper based, web-link, face-to-face meeting) have also been documented in the literature.[2] However, variability of this type is much less problematic (and often warranted) than that for whether the results will be offered at all. It is important to consider that patients from different populations may require different modes of delivery that are appropriate for their needs. The planned changes from the HRA stating they will change the IRAS question from 'whether' results will be disseminated to 'when and how' is welcome but research teams will still require guidance in the what, how and when of dissemination.[5]

Another interesting finding is the similarity between the responses provided in the trial teams applications that intended to provide results to participants and those that did not. Collectively, 72.1% of the applications where trial teams stated they intended to provide results relied on either site staff to provide results or the participants to request the results themselves. Interestingly, these two categories of responses also account for nearly 70% of those applications that responded with 'No' to intention to disseminate results. In certain cases, an identical

**Table 5** Reporting of dissemination of result to trial participants in end of study reports

| Dissemination of results reported | 2013 | 2014 | 2015 | 2016 | 2017 | 2018 | 2019 | TOTAL (n/%) |
|---|---|---|---|---|---|---|---|---|
| No mention | * | 23 | 42 | 58 | 65 | 53 | 35 | 277 (74.9) |
| Confirmation of lay summary/letter* | – | – | – | – | 1 | 3 | 2 | 6 (1.6) |
| Patient end of study sheet attached | – | – | – | – | – | 1 | 1 | 2 (0.5) |
| Follow-up visit | – | – | – | – | – | – | 1 | 1 (0.3) |
| Presentation at scientific conference | – | – | * | – | – | – | – | 1 (0.3) |
| Report inaccessible† | 5 | 3 | 7 | 3 | 16 | 32 | 17 | 83 (22.4) |
| **TOTAL** | 6 | 26 | 50 | 61 | 82 | 89 | 56 | 370 (100) |

*Confirmation of lay summary/letter either as an attached copy or mentioned in final report/cover letter.
†Report inaccessible: Reports that require password/email access or have to be uploaded by the REC.
REC, research ethics committee.

response was provided as justification for intention and no intention. For instance, 'Investigators will be informed of the study results and may pass on the details to participant' was a response that was observed in both the 'yes' and 'no' responses and in some instances was done in applications submitted by the same sponsor. This raises a concern that the question may be interpreted differently by different researchers and that at a conceptual level there is a misunderstanding about what constitutes appropriate methods of disseminating results. Explanatory guidance notes within the IRAS system to ensure how researchers are expected to operationalise and implement the dissemination of results to trial participants may help to resolve some of this lack of continuity.

It is disappointing to see that 69% of the trial teams included in our audit had no intention to include the public at any stage of the research. Nearly 10% of the 850 trials deemed it inappropriate to include the public due to the complexity of the trial. These findings also echo results from an earlier audit of patient involvement in IRAS applications.[11] Particular aspects or types of research may indeed be difficult for a lay person to understand; however, members of the public may still be able to contribute to the participant enrolment or result dissemination phase.[12] A review of publicly funded trials to explore how PPI was included in grant applications identified that most study teams intended to have some form of PPI input.[12] This contrasts with the findings of this audit and others and may reflect the requirement of involvement of patients and/or the public as a condition of funding approval. This raises the question as to whether there could be more linkage between funders to ensure that there is consistency in research teams intentions with regards to involvement and potentially dissemination.

Our study highlights that most End of Study reports do not mention dissemination of results to their participants. However, this may not be surprising given the current guidance is not directive and states, and arrangements for publication or dissemination of the research, including any feedback to participants'.[13] Therefore, more explicit guidance from the Health Research Authority to include information on dissemination of result to trial participants in the End of Study report should be implemented. The changes planned by the HRA as part of their Transparency Agenda will require sponsors to submit a lay summary of the trial results and will attend to aspects of this.[5] This could be strengthened by guidance on what the content of the lay summary should cover and mandating this lay summary as a critical requirement of trial close out

A recent study that surveyed teams that had published trials (involving human participants and enrolling individual patients) during 2014 to 2015 fund only 27% of their eligible sample had disseminated results back to participants with a further 13% planning to do so. This study reported a range of barriers the trial teams identified with regard to disseminating results and summarised these as: researchers perceptions of what interests patients and what they understand, challenges reaching patients, which patients to share with, need for early planning and resource, researcher motivations and situational expectations, type of results to share and researcher specific reasons for not disseminating.[10] The authors propose some helpful suggestions targeting multiple players (such as increased scrutiny from ethics review boards, support from journals and development of standards and training) with the aim of improving practice.

More directive guidance, like the planned changes mentioned above, from the Health Research Authority is required if we plan to change researcher's behaviour with regard to disseminating results of trials to those who participated. The existing guidance published in 2015 does not seem to have impacted on trial teams intentions to disseminate results . Therefore, the current approach of requiring research teams to submit a lay summary at the end of the study seems more appropriate. It would be helpful to go one step further and request sponsor to inform the HRA when and how those results have been offered or disseminated to participants. The participant dissemination activity could be triggered when the trial submits the End of Study report as one of the close out tasks for the team. In addition, the HRA may need to take a more proactive stance with those trial teams who do not submit End of Study reports. Our study has shown that 20.4% of the completed trials either did not submit reports or submitted incomplete data, which is not surprising given there are no consequences for failing to submit.

In addition to the HRA, other stakeholder in the research enterprise could begin to implement systems to ensure dissemination of results to trial participants becomes common place and not, at best, an afterthought. For example, the BMJ pledged in 2019 that they will now ask authors of papers to describe how and when they plan to disseminate findings to research participants.[14]

### Strengths and limitations

This is the first audit of ethics applications reporting trial teams' intention to disseminate results of trials to those who participated. Set within a 5-year time frame we included a large sample (n=1404) of Phase III trials, including a range of clinical populations, interventions, comparators, outcomes and supported by a range of funders. Other IRAS audits have been competed to assess registration of clinical trials given a favourable opinion by UK research ethics committees, which also demonstrated the value of audits of this type to assess current regulatory practice.[15] However, there were limitations to our approach, principally that the IRAS data reports intention and not actual behaviour. There is evidence from health psychology that intention only explain 36% of the variance in behaviour and as such changing intentions does not necessarily engender behaviour change.[16] In other words, the 88% of trial teams reporting they intend to disseminate results will likely be a much lower proportion that actually do it. The other limitation to our study

was that we relied on the identification of phase of trial from trial teams. This may have introduced potential bias in teams misrepresenting their trials or assumptions from our team made with regard to these trials being true pragmatic trials when they may have been nearer the explanatory end of the continuum. Linked to this, it would also be important to consider whether the results of our study are also true for other phases of trials.

## CONCLUSION

According to the HRAs IRAS system, many teams delivering Phase III trials intend to disseminate the results of the trial back to participants. However, reporting of whether this dissemination activity actually happened is much less clear and at best happens in less than half of current Phase III trials approved through IRAS. This isn't surprising given trial teams are not currently mandated to complete End of Study reports and further still there are no specifications on the content of the End of Study reports or any associated lay summaries. There is now potential for this to change with the recent publication of the HRAs transparency agenda but researchers need better guidance on what to report, when and how if the benefits of dissemination are to be realised. Further research is needed to conduct more embedded methodological research in these areas in order to identify best practice about the what, how and when of disseminating trial result to participants.

**Acknowledgements** We thank Bill Davidson (former joint Head of Policy for the Health Research Authority (HRA)) and colleagues at the HRA for enabling access to IRAS data, providing access to HARP and commenting on progress and initial results of the study (specifically, Juliet Tizzard and Jim Elliott). Thanks also to Graeme Maclennan for analysis advice and all members of the RECAP study team (Marion Campbell, Vikki Entwistle, Rosemary Humphreys, Sandra Jayacodi and Peter Knapp).

**Contributors** HB wrote the first draft of the study protocol. KG and MZR contributed to design of the project and development of the protocol. MZR and KG conducted data analysis. All authors (MZR, HB, KG) commented on the results. MZR wrote the first draft of the manuscript. All authors approved the final version of the manuscript.

**Funding** HB was funded by the Academy of Medical Sciences (SBF002\1014) and KG was funded by the Medical Research Council (MR/L01193X/1). MZR was unfunded and conducted this work as part of a Masters in Public Health degree.

**Competing interests** This audit forms part of a larger project that aims to develop recommendations for how to appropriately feedback trial results to those who participated in them. This overarching project is funded by the Academy of Medical Science (SBF002\1014: Chief Investigator KG and supported HB, RECAP: researchregistry4085). KG was supported by an MRC Methodology Research Fellowship (MR/L01193X/1).

**Patient consent for publication** Not required.

**Provenance and peer review** Not commissioned; externally peer reviewed.

**Data availability statement** Data may be obtained from a third party and are not publicly available. Data requests should be made to the Heath Research Authority.

**ORCID iD**
Katie Gillies http://orcid.org/0000-0001-7890-2854

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
