## [Reviewer comments · BMJ Open]

ARTICLE DETAILS

TITLE (PROVISIONAL)	Dissemination of trial results to participants in Phase III pragmatic clinical trials: an audit of trial investigators intentions.
AUTHORS	Raza, M.Zulfiqar; Bruhn, Hanne; Gillies, Katie

VERSION 1 - REVIEW

REVIEWER	Simon Kolstoe University of Portsmouth, UK I chair an HRA research ethics committee, sit on the HRA Transparency Steering group, and conducted an audit using a similar method (but on a different topic) that has been referenced by these authors. I conduct training for the HRA, UKRIO and ARMA that covers the topic described in this paper.
REVIEW RETURNED	22-Nov-2019

GENERAL COMMENTS	This is a valuable audit that adds to the important ongoing discussion concerning research transparency and specifically reporting results back to participants. While this is the first study to use IRAS records to audit dissemination practice, the method used by these authors has been validated by a previous paper published in this journal. The results and discussion are important because, while perhaps not surprising, they do provide solid data to back up what has hitherto been an anecdotal argument regarding dissemination practices. The justification of limiting this study to phase III trials is pragmatic and sensible. My only condition prior to publication would be to reconsider figure 2. It is currently uninformative and I suggest either deleting it entirely or reformatting is as a scatter plot showing confidence intervals.
---

REVIEWER	Satish Chandra Nair Tawam Hospital Johns Hopkins Medicine Affiliate, United Arab Emirates
REVIEW RETURNED	26-Nov-2019

GENERAL COMMENTS	I commend the authors for an interesting study. In order to strengthen the manuscript, please refer to the points stated below: 1) The premise of pragmatic trials is the fact that all patients who are potential candidates for treatments in routine clinical practice
--

	are eligible for the trial, without adding burden to the investigator and the patient, with no compromise on clinical adherence. 2) It would be ideal if the authors categorize their observations based on study duration and the therapeutic areas. 3) Dissemination of trial results to participants in case of pragmatic trials can be challenging for the lack of a specific milestone at the end of the trial to trigger the dissemination pathway for trial results to participants. This is in contrast to the exploratory trials where a milestone such as a US-FDA approval can trigger many complex pathways. The authors need to clarify if there is a milestone for pragmatic trials developed in the UK, that can trigger the dissemination pathway. 4) The statistical representation could have been better with the percentage of concordance and partial concordance be compared with the discordance, quite similar to a McNemar test. 5) It is not clear from the manuscript whether “dissemination of pragmatic trial results” to participants is a mandatory regulatory requirement by the ethics committee and regulatory bodies. 6) Are the investigators and research ethics committee members trained to understand the complex pathways of results dissemination of trial results and the consequences thereafter? 7) How was the “intent to disseminate” versus “actively disseminate” differentiated for the study? This is relevant given the fact that there is not an accepted framework for active dissemination of trial results 8) Interestingly, in the light of the above, the authors report that the “End of Study report” was submitted by only 59.4 % of completed trials, indicating the lack of a prescribed framework. 9) The highlight of the manuscript is the public involvement in the trial, part which addresses a significant point, although nearly 70% of the studies had no intention to include the public at all. 10) The points listed above from 1-9 needs to be addressed clearly by the authors before the manuscript can be considered for publication
--	---

REVIEWER	Sara Schroter BMJ
REVIEW RETURNED	29-Nov-2019

GENERAL COMMENTS	Dissemination of trial results to participants in Phase III pragmatic clinical trials: an audit of trial investigators intentions I read this paper with interest. It is an important and original contribution to the literature on dissemination as it illustrates some important gaps between what researchers report they intend to do and actual practice. I have a number of minor comments or clarifications. General comment Throughout the manuscript there are statements like “Proportion of trials that intended....”(L16, L20). Trials don’t have intentions, researchers/authors do, nor do trials report at end of a study (L17). Abstract L14: Will international readers know what IRAS is?
--

L17: Perhaps indicate that reporting at the end of study meant the report for the ethics committee. Not immediately clear if you mean journal publications.

L19-21: These do not sum to 1231 - did the rest not answer this question?

L21-23: while interesting, this is not the key focus of the study and probably doesn't need to be in the abstract. L23-25 are more important and need a bit more detail.

L23: perhaps say where these were accessed from? Add the proportion of all the included studies (30%). 10 of 370 = 2.7% mentioned dissemination and 74.9% had no mention, what happened to the rest? These do not sum to 100%.

L26: This should say "Reported intention to..."

L27: The meaning of "reporting of appropriate feedback methods is lacking" is not clear.

Article summary

L34: This should say "investigators reported intention to..."

Introduction

L46: Spell out NIHR CRN for non UK readers.

L45-50: The fact that there is more research being done isn't the key reason behind the need for improved transparency and PPI. I think this paragraph could be strengthened.

L59: explain what the HRA is for non-UK readers.

Methods

L105-106: needs rephrasing.

L143-144: say how these patient partners have contributed to this study.

Results

L151: state that these 6826 trials had a favourable REC decision.

Figure 1A: Indicated in the box "Other phase trials" that these were excluded.

Figure 1A: Change boxes to "Reported intention to disseminate..." and "No reported intention to disseminate..."

Figure 1A: the last 2 boxes are not key to the sampling strategy and should be removed. It would be more helpful to indicate availability of End Study Report here.

Figure 1B: This pie chart should be removed - it doesn't add anything that isn't in the text.

L168: actually the Nos also responded so remove this part about "if yes"....

L172: Mention Table 1 here rather than the end of the paragraph so the reader knows they can see all this in a table.

L176: "74/disseminate" appears to be a typo.

Table 1: the meaning of all items listed is not clear eg "Trial linked staff" - please add some more detail to each line in the table.

L205-13: Other factors could have led to these differences. It is not clear how many studies were in the pre and post samples and whether the sample size was sufficient to show a change. I would remove the significance testing.

L227-234: be careful not to use past tense eg L230 "had input", L231 "had input", L232 "involved".

L265: I wouldn't describe 42% as "several".

L265-271: it would be easier to read if you removed those still in the 12month post study period and then reported how many had submitted an end of study report versus had not. Of the 497 completed trials that were outside of the 12 month period, 74% had submitted a report. This is a lot higher than 59%. It would be good if the Discussion section picked this up - why are RECs not requesting the reports from the missing 25%? What can be done to meet this gap? Is this the same subset referred to in L298? Why are there only 83 in L298?

Table 5: I am not sure that presenting the data for each year adds anything, the total across all years could just be reported.

Discussion

L319: Again, it would be more accurate to report this after excluding those in the 12 month post study period, ie the 74%.

L324: "with several trial teams not doing so" - there were many more than several.

L326: variability in methods could be a good thing with various methods of dissemination for different patient groups and populations. There is no one size fits all...

L347- 357: Does this indicate a training need or improved resources so that researchers become more familiar with how to do PPI?

L362-364: You may also want to mention somewhere in the Discussion that journals can also play a role in encouraging dissemination to participants. At The BMJ we now ask authors of accepted research papers to describe plans for dissemination of their findings to research participants and other relevant communities. We want to know how patients and the public were, or will be, involved in choosing the methods and developing plans to share research findings, and when and how dissemination has been or will be done.

(<https://www.bmj.com/content/364/bmj.k5428>)

L365: perhaps mention if it will be mandatory to make the lay summary publicly available?

L387-399: another limitation is the focus only on phase 3 trials. How do you expect the results to have differed if you broadened the inclusion criteria?

	L398: what impact do you think this had on your results? Conclusion L402-403: This could be phrased more clearly to make the key points. Many people just read the conclusion. Compliance with reporting what is requested from researchers is actually quite high on IRAS - researchers are not requested to mention dissemination in the end of study report so it is not surprising that they don't. Researchers need better guidance on what to report and when and there needs to be a better system for checking that they do report these things. Further research is needed to find out if this is just a reporting problem or if researchers are not disseminating to participants as they reported they intended to. It is beyond the scope of this study but it would be interesting to contact the researchers and ask them if they disseminated and if not why not? L411-421: You have not acknowledged or thanked your patient partners for their contributions to this research.
--	---

VERSION 1 – AUTHOR RESPONSE

Reviewer #1	
1	This is a valuable audit that adds to the important ongoing discussion concerning research transparency and specifically reporting results back to participants. While this is the first study to use IRAS records to audit dissemination practice, the method used by these authors has been validated by a previous paper published in this journal. The results and discussion are important because, while perhaps not surprising, they do provide solid data to back up what has hitherto been an anecdotal argument regarding dissemination practices. The justification of limiting this study to phase III trials is pragmatic and sensible. My only condition prior to publication would be to reconsider figure 2. It is currently uninformative and I suggest either deleting it entirely or reformatting is as a scatter plot showing confidence intervals. Response Given additional comments from other reviewers about the added value of this figure we have chosen to delete it and linked text in Methods, Results, and Discussion section. We have left the statement within the discussion section (line328) that indicates the existing guidance did not seem to influence intentions but referred to unpublished data. This point helps supports the need for additional approaches to ensure the reporting of results to participants highlighted in the proceeding sentence.
Reviewer #2	
1	The premise of pragmatic trials is the fact that all patients who are potential candidates for treatments in routine clinical practice are eligible for the trial, without adding burden to the investigator and the patient, with no compromise on clinical adherence. Response We agree that that one aspect of pragmatic trials is that they replicate real world practice as much as possible. The statements we make on lines 100-104 (which we assume is the section

	to which the reviewer is making this statement) are to highlight the participant-facing reasons why investigators of Phase III (often pragmatic) trials should critically consider the need to disseminate result to those who contributed.
2	It would be ideal if the authors categorize their observations based on study duration and the therapeutic areas. Response We recognise there could be additional insights gleaned from the data by analysing investigator intentions based on clinical area and/or duration. However, this data either cannot be obtained easily (therapeutic area) or at all (duration) from the IRAS data provided. Therefore, we are unable to conduct these further analyses.
3	Dissemination of trial results to participants in case of pragmatic trials can be challenging for the lack of a specific milestone at the end of the trial to trigger the dissemination pathway for trial results to participants. This is in contrast to the exploratory trials where a milestone such as a US-FDA approval can trigger many complex pathways. The authors need to clarify if there is a milestone for pragmatic trials developed in the UK, that can trigger the dissemination pathway. Response We had implicitly suggested this 'dissemination pathway' but have now included a sentence to make it clearer that the milestone that could trigger the activity would be the submission of the End of Study report to the HRA. See lines 364-366.
4	The statistical representation could have been better with the percentage of concordance and partial concordance be compared with the discordance, quite similar to a McNemar test. Response As per the response to reviewer 1 above we have decided to remove this figure and the text within the relevant sections of the manuscript.
5	It is not clear from the manuscript whether "dissemination of pragmatic trial results" to participants is a mandatory regulatory requirement by the ethics committee and regulatory bodies Response We have now included a sentence within the introduction to highlight to the reader that dissemination of results to participants is not a mandatory requirement and contrast this with PPI which is mandated. Lines 79-81.
6	Are the investigators and research ethics committee members trained to understand the complex pathways of results dissemination of trial results and the consequences thereafter? Response

	No. Currently there is no standard training provided to investigators or RECs to provide support in dissemination of trial results. However, we believe this is outside the scope of our article. In addition, specific aspects of potential training for trial teams is covered by a linked article published recently in BMJ open to which we reference in our paper (Schroter S, et al BMJ Open. 2019 Oct 21;9(10): e032701)..
7	How was the “intent to disseminate” versus “actively disseminate” differentiated for the study? This is relevant given the fact that there is not an accepted framework for active dissemination of trial results Response The intention of trial teams to disseminate to participants was identified from their response (of ‘Yes’ or ‘No’) to a question within the IRAS form asking whether they would inform participants of the results (A53 – See Box 1). The IRAS form then requests trial teams to provide further information on the details of the plans if they intended to disseminate but also to justify if not doing so. This free text data was analysed using a content analysis approach which resulted in our team coding practices as active or passive. We have edited the methods section to try to make this process clearer. Lines 144-150.
8	Interestingly, in the light of the above, the authors report that the “End of Study report” was submitted by only 59.4 % of completed trials, indicating the lack of a prescribed framework. Response No response required.
9	The highlight of the manuscript is the public involvement in the trial, part which addresses a significant point, although nearly 70% of the studies had no intention to include the public at all. Response The findings from analysis of the public involvement in key aspects of trial design and delivery was a secondary outcome of the study. Whilst interesting we do not wish to make this the focus of the manuscript.
Reviewer #3	
1	I read this paper with interest. It is an important and original contribution to the literature on dissemination as it illustrates some important gaps between what researchers report they intend to do and actual practice. I have a number of minor comments or clarifications. General comment Throughout the manuscript there are statements like “Proportion of trials that intended...”(L16, L20). Trials don’t have intentions, researchers/authors do, nor do trials report at end of a study (L17). Response Edited throughout.

2	Abstract L14: Will international readers know what IRAS is? Response We have changed the text in the abstract (Line 14) to be more descriptive and also included a sentence within the Methods section (lines 118-119) to provide more explanation about what IRAS is.
3	L17: Perhaps indicate that reporting at the end of study meant the report for the ethics committee. Not immediately clear if you mean journal publications. Response Edited accordingly – line 17
4	L19-21: These do not sum to 1231 - did the rest not answer this question? Response This is an error in the manuscript. The abstract has been edited to reflect the text in the results section and tables. It should read 991 rather than 990 and indicate a further 9 studies could not be categorised. Please now see amended text on lines 21-22.
5	L21-23: while interesting, this is not the key focus of the study and probably doesn't need to be in the abstract. L23-25 are more important and need a bit more detail. Response We have removed the text on involvement of patient partners from the abstract and included further information on the data regarding End of Study Reports. Lines 22-28.
6	L23: perhaps say where these were accessed from? Add the proportion of all the included studies (30%). 10 of 370 = 2.7% mentioned dissemination and 74.9% had no mention, what happened to the rest? These do not sum to 100%. Response In the interests of brevity for the abstract we don't believe it is a requirement to include how and where the reports were accessed from. This information is provided in the methods. We have edited the remainder of the results section of the abstract to allow details of the findings to be clearer in the text. Lines 26-28.
7	L26: This should say "Reported intention to...." Response Edited accordingly
8	L27: The meaning of "reporting of appropriate feedback methods is lacking" is not clear.

	Response We have removed the word appropriate from this sentence to make the statement clearer and not require explanation within the abstract. Line 30.
9	
10	Article summary L34: This should say “investigators reported intention to....” Response Edited accordingly
11	Introduction L46: Spell out NIHR CRN for non UK readers. Response Edited accordingly
12	L45-50: The fact that there is more research being done isn't the key reason behind the need for improved transparency and PPI. I think this paragraph could be strengthened. Response Opening section of the introduction has been edited to strengthen content and readability. Lines 52-56 and 64-68.
13	L59: explain what the HRA is for non-UK readers. Response Edited accordingly
14	Methods L105-106: needs rephrasing. Response Edited accordingly
15	L143-144: say how these patient partners have contributed to this study. Response Edited accordingly
16	Results L151: state that these 6826 trials had a favourable REC decision. Response

	Edited accordingly
17	Figure 1A: Indicated in the box “Other phase trials” that these were excluded. Response Edited accordingly
18	Figure 1A: Change boxes to “Reported intention to disseminate....” and “No reported intention to disseminate...” Response Edited accordingly
19	Figure 1A: the last 2 boxes are not key to the sampling strategy and should be removed. It would be more helpful to indicate availability of End Study Report here. Response Edited accordingly
20	Figure 1B: This pie chart should be removed - it doesn't add anything that isn't in the text. Response Edited accordingly
21	L168: actually the Nos also responded so remove this part about “if yes”.... Response Edited accordingly
22	L172: Mention Table 1 here rather than the end of the paragraph so the reader knows they can see all this in a table. Response Edited accordingly
23	L176: “74/disseminate” appears to be a typo. Response Edited accordingly

24	Table 1: the meaning of all items listed is not clear eg "Trial linked staff" - please add some more detail to each line in the table. Response The table has now been edited to add more detail which complements the text provided in the results section of the manuscript.
25	L205-13: Other factors could have led to these differences. It is not clear how many studies were in the pre and post samples and whether the sample size was sufficient to show a change. I would remove the significance testing. Response As per comments from previous reviewers this analysis and the figure have been removed from the manuscript.
26	L227-234: be careful not to use past tense eg L230 "had input", L231 "had input", L232 "involved". Response This section has now been edited to ensure past tense is avoided and trial teams rather than trials reported the intentions
27	L265: I wouldn't describe 42% as "several". Response We have edited this to state 'A large proportion' rather than several/ line 262.
28	L265-271: it would be easier to read if you removed those still in the 12month post study period and then reported how many had submitted an end of study report versus had not. Of the 497 completed trials that were outside of the 12 month period, 74% had submitted a report. This is a lot higher than 59%. It would be good if the Discussion section picked this up - why are RECs not requesting the reports from the missing 25%? What can be done to meet this gap? Is this the same subset referred to in L298? Why are there only 83 in L298? Response This comment from the reviewer highlighted a mistake in the text of the manuscript which may have led to misunderstanding of the data presented and the request for edits. As per the text in Table 4, the text in the manuscript should have detailed the final 10.3% as trials with incomplete data registered on HARP and as such made inclusion in analysis difficult. This has now been corrected in the manuscript text (lines 267-268). Therefore the analysis as presented is correct due to all of the 624 reports contained within the completed trials should have reported an End of Study report to HARP as they are out with the 12 month completion timeline.

	However, the point regarding why RECs are not requesting the data from the trial teams for which it is currently missing (20.4%) is a good suggestion and has been included in the discussion. Lines 367-369. The sample of 83 reported in the last sentences the results section are a different sample. These are the trials for whom they had indicated an end of study report had been uploaded (part of the 370) but then the reports could not be accessed. Further details in the text have now been included to be more explicit. Lines 277.
29	Table 5: I am not sure that presenting the data for each year adds anything, the total across all years could just be reported. Response We agree that presenting the findings by year don't add anything additional, however, we feel it is still important to present the data by year given the hypothesis that reporting may have improved over time. This allows the reader to see that this is currently not the case.
30	Discussion L319: Again, it would be more accurate to report this after excluding those in the 12 month post study period, ie the 74%. Response As per response to earlier point (#28) this data is correctly presented but due to an error in the text it had been misinterpreted.
31	L324: "with several trial teams not doing so" - there were many more than several. Response We have edited from 'several' to 'many' – line 299.
32	L326: variability in methods could be a good thing with various methods of dissemination for different patient groups and populations. There is no one size fits all... Response We agree and already include a statement to reflect this in the existing text but have now expanded this to make it more explicit. Lines 303-304.
33	L347- 357: Does this indicate a training need or improved resources so that researchers become more familiar with how to do PPI? Response It may do but as the focus of this paper is on dissemination of results to participants we would like this paragraph to close out considering the link between PPI and dissemination, as it stands, rather than PPI more broadly.

34	L362-364: You may also want to mention somewhere in the Discussion that journals can also play a role in encouraging dissemination to participants. At The BMJ we now ask authors of accepted research papers to describe plans for dissemination of their findings to research participants and other relevant communities. We want to know how patients and the public were, or will be, involved in choosing the methods and developing plans to share research findings, and when and how dissemination has been or will be done. (https://www.bmj.com/content/364/bmj.k5428) Response Thank you to the reviewer for providing this helpful reference. We have now included a few sentences to acknowledge the role that journals could play in ensuring the process of dissemination happens. Lines 371-375.
35	L365: perhaps mention if it will be mandatory to make the lay summary publicly available? Response We have now included text to highlight to the reader that while not currently mandated, provision of lay summary could be considered a mandatory requirement of trial close out-Lines 345
36	L387-399: another limitation is the focus only on phase 3 trials. How do you expect the results to have differed if you broadened the inclusion criteria? Response We have now included text to address this point -Lines 391 - 393.
37	L398: what impact do you think this had on your results? Response We have now included text to address this point -Lines 389-391.
38	Conclusion L402-403: This could be phrased more clearly to make the key points. Many people just read the conclusion. Compliance with reporting what is requested from researchers is actually quite high on IRAS - researchers are not requested to mention dissemination in the end of study report so it is not surprising that they don't. Researchers need better guidance on what to report and when and there needs to be a better system for checking that they do report these things. Further research is needed to find out if this is just a reporting problem or if researchers are not disseminating to participants as they reported they intended to. It is beyond the scope of this study but it would be interesting to contact the researchers and ask them if they disseminated and if not why not? Response The conclusion has now been edited. Lines 396-411.
39	L411-421: You have not acknowledged or thanked your patient partners for their contributions to this research.

	Response We have now extended a sentence in the acknowledgements section to recognise all member of the RECAP study team, which includes patient partners. Other patient partner, Jim Elliot, is already acknowledged.

VERSION 2 – REVIEW

REVIEWER	Satish Chandra Nair Tawam Hospital Johns Hopkins Med
REVIEW RETURNED	19-Dec-2019

GENERAL COMMENTS	The reviewer completed the checklist but made no further comments.
--

REVIEWER	Sara Schroter BMJ
REVIEW RETURNED	18-Dec-2019

GENERAL COMMENTS	The authors have responded satisfactorily to all my comments. However lines 407-409 need to be revised - BMJ doesn't ask for evidence of dissemination to participants but when and how dissemination has been or will be done.
---

VERSION 2 – AUTHOR RESPONSE

The author provided a marked copy with additional comments. Please contact the publisher for full details.